# Elderly Health Inequality in China and its Determinants: A Geographical Perspective

**DOI:** 10.3390/ijerph16162953

**Published:** 2019-08-16

**Authors:** Chenjing Fan, Wei Ouyang, Li Tian, Yan Song, Wensheng Miao

**Affiliations:** 1School of Architecture, Tsinghua University, Beijing 100084, China; 2College of Landscape Architecture, Nanjing Forestry University, Nanjing 210037, China; 3School of Public Administration, Renmin University of China, Beijing 100872, China; 4Department of City and Regional Planning, University of North Carolina at Chapel Hill, NC 27599-3140, USA; 5China Research Center on Aging, Beijing 100054, China

**Keywords:** Elderly health, health inequality, geographical differentiation, multilevel regression

## Abstract

Inter-regional health differences and apparent inequalities in China have recently received significant attention. By collecting health status data and individual socio-economic information from the 2015 fourth sampling survey of the elderly population in China (4th SSEP), this paper uses the geographical differentiation index to reveal the spatial differentiation of health inequality among Chinese provinces. We test the determinants of inequalities by multilevel regression models at the provincial and individual levels, and find three main conclusions: 1) There were significant health differences on an inter-provincial level. For example, provinces with a very good or good health rating formed a good health hot-spot region in the Yangtze River Delta, versus elderly people living in Gansu and Hainan provinces, who had a poor health status. 2) Nearly 2.4% of the health differences in the elderly population were caused by inter-provincial inequalities; access (or lack of access) to economic, medical and educational resources was the main reason for health inequalities. 3) At the individual level, inequalities in annual income served to deepen elderly health differences, and elderly living in less developed areas were more vulnerable to urban vs. rural-related health inequalities.

## 1. Introduction

Health inequalities are differences in health that are deemed to be avoidable and unjust, and such inequalities can be revealed through observed patterns of health outcomes across populations [1]. The subject of health inequalities has attracted the attention of scholars in public health, and numerous studies have shown that poor people might suffer from unequal social resources, and these inequalities are manifested in poor health outcomes [2]. Taking China as an example, we find that the health status of many citizens has improved along with the country’s rapid economic development in recent years. However, widening health inequalities have also been observed. Some studies have pointed out that high-income residents have better health than low-income groups, controlling for demographic characteristics such as gender, age and education [3,4,5].

Reducing health inequalities is a result of social progress and may have a profound impact on the future development of a society [6]. Recently, geographic concerns have been increasingly involved with public health. From the perspective of geography and economics, we can better understand inequalities in the health levels of different populations. Curtis (2004) [6] constructed a set of conceptual frameworks that synthesized the relationships between health and different factors such as health care, social policy, economics, consumption and ecological environment, all of which may have a strong impact on health. Curtis also illustrated the intricate intertwining of spatial variations and inequalities by using a large number of case studies from various populations around the world.

Given the spatial dimension of health inequities, in this paper, we analyze health indicators geographically, using approaches afforded by geospatial analysis; namely, conducting a geographic analysis of health indicators that reveals the spatial distribution of geographic areas, identifying areas that require focused attention and improving the precise allocation of resources and interventions in areas in which they are most needed [7,8,9,10]. A series of empirical studies have shown that environmental exposure, and the unequal distribution of medical resources influenced by geographic considerations have caused public health problems [11]. Some areas were more susceptible to toxic and hazardous substances, and people living in these areas were vulnerable to unequal treatment [12,13,14,15,16,17,18]. Also, differences in the allocation of green space and medical resources may generate different impacts on the health levels of different people. The former studies have tested various hypotheses regarding the effects of circumstances on the health of the population. For example, Pearce and Dorling (2006) and Pearce et al. (2010) conducted research on health inequalities in different regions and found that regions in the UK and New Zealand with low income levels tended to have poor health [19,20,21,22]. Wallace (2017) showed that education and ethnic inequality among different socio-economic status, led to different infant mortality rates in the United States [23]. In China, recent studies have observed the phenomenon of health differences between the eastern and western regions [24], as well as at the interprovincial level [25,26,27] and between rural and urban areas [5,28,29] accompanied by rising economic disparities. Meanwhile, the health status of different regions varies considerably, depending on the social, economic, and environmental conditions of regions; therefore, the unequal distribution of medical facilities in China is addressed in this paper by means of a study of social and public health resources. Zhang (2017) and Zhao (2018) pointed out that economically developed regions often have better medical facilities [30,31]; however, the question of whether the unequal distribution of resources in China leads to health differences has not been sufficiently discussed.

Understanding health variations from a geographical perspective and developing a viable cultural, economic and social policy to ameliorate inequalities will have valuable future implications for regions that are currently less developed but are rapidly urbanizing [6]. Compared with other populations, the elderly are less physically active and have reduced self-healing abilities, and their health is more susceptible to socio-economic and environmental changes. Recently, researchers in China have carried out a large number of studies on the health status of the elderly [32]. For instance, individual-level research indicates that income, education, loneliness and housing conditions can affect the health of elderly people [33,34,35]. Ruan et al. (2017) found that an imbalance in medical resources and the living environment cause health inequalities among the elderly in Beijing [35,36]. Du et al. (2013) analyzed the provincial differences of the health status of the elderly in China and found that the elderly in the eastern region were healthier than in other regions [37]. Additionally, there are more pollution emissions in regions with high development in China, and an empirical study by Sun (2008) found that elderly people who live in more developed regions were more susceptible to the effects of air pollution than those who live in less developed regions [38].

In general, there is a complex and intertwined relationship between inequalities in resource distribution and health differences. Given China’s diversity, variety and regional disparities, research on health inequalities has recently increased. Current research on elderly health focuses on income-related inequalities, and research on the impacts of environment, education and medical resources on health inequalities have been fairly scarce among provinces. In this research, we used self-assessment health data and provincial spatial information to observe the geographical differentiation of Chinese elderly health based on the fourth sampling survey of the elderly population (4th SSEP) in China, which was conducted in 2015 by the National Aging Committee. Based on previous research on health inequalities, we hypothesize that regional health disparities are connected by differences in socio-economic situations, medical care [30,31], education [23] and inequalities in environmental quality [36,38,39,40,41,42] at the provincial level. Meanwhile, individual characteristics such as income and living environment [35] also influence health equality. A cross-sectional study that employs data from the 4th SSEP survey, and includes a number of multi-level linear regression models, was constructed to explore whether health differences in China are related to these geographically different factors at the provincial and individual levels.

## 2. Methods

### 2.1. Data

#### 2.1.1. Individual-Level Variables

Since the number of elderly people in China is increasing, an aging society is developing. The China Research Center on Aging (CRCA) provided the 4th SSEP data for 222,179 people over 60 years old in 2015, which was the first 1‰ sampling survey of the elderly (sample survey of a population over 60 years old) in China. The goal of this survey was to investigate the living environment of elderly people in China and evaluate the socio-economic, psychological and physical health status of elderly people in different regions. Our work included geographical identifiers for 31 provinces (excluding Hong Kong, Macao and Taiwan) and residential locations (urban, town or rural). All subjects gave their informed consent for inclusion before they participated in the survey.

#### 2.1.2. Provincial-Level Variables

At the provincial level, we had collected economic, medical, educational and environmental data in the areas where the elderly population lived. We focused on factors that may affect health equality/inequality according to Ruan et al. (2017) [35] and Curtis (2004) [6], and climate-related factors such as temperature, rainfall, and sunshine were also considered [43,44,45]. The economic, medical resource, educational and environmental data came from the “The Sixth Population Census,” and the “China Urban Statistical Yearbook” in 2015. For air pollution data, research had shown that air pollution could yield significant mid-term effects on health, and air pollution indexes for the year of 2010 were used considering its lagging influence on health [38]. This data was collected from the “Datacenter in Ministry of Environmental Protection of China”. Table 1 and Figure 1 present these variables, and the descriptive statistics of individual-level and provincial-level variables are shown in Table 2.

### 2.2. Ill-Health Score

The dependent variable in this research is self-assessed health. Previous studies suggest that this is a sensitive and reliable indicator of current health status with high predictive validity independent of other medical, behavioral or psycho-social factors [46]. In the 4th SSEP data, the health variables of the elderly were set to a five-category self-assessed ordering of variables “very good, good, fair, poor, very poor”. In this article, we transformed the self-assessed ordering of health to an ill-health score using the method used by Wagstaff et al. [26,35,47]. We assumed that underneath the categorical self-assessed health variable is a continuous variable representing the individual's self-assessed health, and we noted the latent health variable that had a standard normal distribution. Then, according to the proportion of the five-category self-assessed ordering variables, the five-category ill-health score *Y* could be produced according to the standard normal distribution (Equation 1):(1)$${\hat{Y}}_{c}=\left(\frac{N}{n_{c}}\right)\left[\Phi (\Phi^{-1}\left(\overset{c-1}{\underset{i=1}{\sum }}\frac{n_{i}}{N}\right))-\Phi \left(\Phi^{-1}\left(\overset{c}{\underset{i=1}{\sum }}\frac{n_{i}}{N}\right)\right)\right]$$ where $\Phi^{-1}$ is the inverse standard normal cumulative density function; $n_{i}$ is the number of cases in category *c* (very good = 1, good = 2, fair = 3, poor = 4, very poor = 5); and *N* is the total number of cases. ${\hat{Y}}_{c}$ is the normal score for category *c*, and Φ is the standard normal density function. Using this method, a higher ill-health score *Y* means poorer health.

### 2.3. Spatial Autocorrelation of Elderly Health

The spatial autocorrelation analysis method is a data mining method used to help understand the degree to which one object is similar to other nearby objects. We first used the global autocorrelation method to derive the spatial patterns of health status at the provincial level by means of Moran's *I* index [48]. Subsequently, we used the local spatial autocorrelation method to measure the association of health status in each province with its neighboring provinces and adopt Local Moran's *I* index to identify the specific spatial agglomeration pattern [49,50,51]. The formula is as follows:(2)$$Local\;Moran^{'}s\;I=\;\frac{n\left(Y_{i}-\bar{Y}\right)\sum_{j=1}^{m}W_{ij}\left(Y_{j}-\bar{Y}\right)}{\sum_{i=1}^{n}{\left(Y_{i}-\bar{Y}\right)}^{2}},\;\left(i\neq j\right)$$ where $Y_{i}$ and $Y_{j}$ are the average ill-health scores of province *i* and province *j*; *n* is the number of provinces; $W_{ij}$ is the spatial weight matrix, which is established based on the common side or common point of each province using the queen contiguity adjacency standard—i.e., when province *i* and *j* are adjacent, $W_{ij}$ = 1, otherwise $W_{ij}\;$= 0; *i* = 1, 2,...,*n*; *j* = 1,2,...,*m*; *m* is the number of provinces adjacent to province *i*; and $\bar{Y}$ is the average of the ill-health score. The significance level of the Local Moran’s *I* index is measured by Z(I)*,* and the significance threshold is 1.96. When Local Moran’s *I* > 0 and Z(I) > 1.96, the province has a higher ill-health score value, a higher average of neighboring values above the mean, and the formation of a poor health hot-spot region exists. When Local Moran’s *I* > 0 and Z(I) < −1.96, the province is a good health hot-spot region.

### 2.4. Health Concentration Curve 

By means of the ill-health scores, we identified the spatial difference variable in the distribution of socio-economic status (SES) at the provincial and individual levels. The concentration curve plots the cumulative proportion of ill-health scores *Y* (on the vertical axis) against the cumulative proportion of the sample (on the horizontal axis), ranked by income, medical resources, education, environment (or some other measure of SES inequality), beginning with the most disadvantaged province or person. If the concentration curve lies above the diagonal, the ill-health score is typically among the worst. The further the curve lies from the diagonal, the greater the degree of inequality in the SES distribution. Further, the concentration index *CI* is defined as twice the area between the curve and the diagonal, and can be written as below:(3)$$CI=\;\frac{2}{\bar{Y}n}\overset{n}{\underset{i=1}{\sum }}\left(Y_{i}*R_{i}\right)-1$$ where *R_i_* is the fractional rank of the *i*th person in the income, medical resources, education, environmental (or some other measure of SES) distribution equal to $R_{i}=\;\frac{2i-1}{2n}$; and *Y_i_* is the ill-health score of the *i*th person. *CI* is a measure of relative inequality, and the value of *CI* is between –1 and 1. When *CI* < 0, this indicates that the lower the value of an SES factor, the greater the number of poor healthy elderly people. Taking *CI* for the income-related ill-health score inequality as an example, a *CI* below zero means that the people with low income are in poor health [2,47].

### 2.5. Determinants of Health Inequality

In the past, researchers were aware of problems due to the nesting of individuals within higher-level units of data hierarchy; therefore, the presence of similarities among individuals in the same groups should not enter directly into single-level analyses. Failure to account for similarities among individuals in the same region might lead to biased estimates of model parameters and produce erroneous conclusions about the effects of certain predictors in the model [52]. Multi-level regression modeling represents a compromise between modeling each unit separately and modeling all unit contexts simultaneously within the same model [53].

In this study, the difference in health status caused by socio-economic differences at the provincial level is our primary concern; however, the individual-level factors cannot be ignored. Such factors are responsible for most of the health differences that are found by researchers who use traditional methods. Thus, multi-level regression modeling rather than single level regression modeling must be used. In this study, a two-level regression model that includes provincial and individual level factors was used to study the determinants of health differences. There are three distinct steps involved in developing the multilevel regression model. The first step is to develop a null model to partition the variance in the outcome into within-groups and between-groups components. This helps determine how much of the variance in ill-health score *Y* lies between the provinces in the sample [54]. The null model for individual elderly *i* in province *j* is shown in Equation (4):(4)$$Y_{ij}=B_{0j}+\varepsilon_{ij}$$ where $B_{0j}$ is the intercept and $\varepsilon_{ij}$ represents variation in estimating the individual health status within the provinces. Among provinces, the variation in intercepts can be represented as Equation (5):(5)$$\beta_{0j}=\gamma_{00}+u_{0j}$$

Through substitution, the null model can be written as Equation (6):(6)$$Y_{ij}=\gamma_{00}+u_{0j}+\varepsilon_{ij}$$

The null model therefore provides an estimated mean health status *Y* for all provinces. It also provides a partitioning of the variance between Level 1 individual-level ($\varepsilon_{ij}$) and Level 2 provincial-level ($u_{0j}$). The intraclass correlation coefficients (ICC) describe the proportion of variance that is common to each unit, and it determines whether the health difference of the elderly population will be significantly affected by the inter-provincial geographical differentiation. If there is a significant difference, the provincial variables will be introduced on the basis of Equation (6), and Equation (7) will be further constructed to determine the influence of provincial-level factors on elderly health. At the same time, individual-level variables are added to Equation (7), and a multi-level regression model of elderly population health (Equation (8)) is constructed to test the SES inequalities determinants of elderly health differences at the provincial-level:(7)$$\beta_{0j}=\gamma_{00}+\gamma_{01}S_{1}+\gamma_{02}S_{2}+\gamma_{03}S_{3}+\gamma_{04}S_{4}+\gamma_{05}S_{5}+\ldots +u_{0j}$$
(8)$$Y_{ij}=\beta_{0j}+\gamma_{10}X_{1}+\gamma_{20}X_{2}+\gamma_{30}X_{3}+\gamma_{40}X_{4}+\ldots \varepsilon_{ij}$$ where $S_{j}\;$is a provincial–level variable, $X_{n}$ is an individual-level variable, and the variable calculation method is shown in Table 2. In the provincial-level data, we focus on the inequality in regional economic, educational, environmental and medical resource disparities—and their effect on elderly health among provinces. In the multi-level regression of Equation (8), we construct five models of health inequalities, focusing on the health effects of factors such as individual characteristics, location (urban vs. rural), and individual economic situations.

## 3. Results

### 3.1. Spatial Pattern of Elderly Health

The descriptive statistics of the ill-health score are shown in Table 2 according to Equation (1), and the average ill-health score *Y* in 31 provinces is calculated based on the geographical location information of the provinces (Figure 1), and five groups—very good, good, fair, poor, very poor —were divided by a natural break method according to the ill-health score. The very poor health group had an average ill-health score greater than 1.901, mainly located in the provinces of Hainan and Gansu. The scores of poor health, fair, and good health groups range from 1.761–1.900 in 3 provinces, 1.541–1.760 in 14 provinces, and 1.271–1.540 in 6 provinces, respectively. The ill-health score of the very good health group was lower than 1.270, and this group was located in Beijing, Tianjin, Jiangsu, Shanghai, Zhejiang, and Fujian (Figure 2).

The global Moran's *I* index of China's elderly ill-health score was 0.207 at the provincial-level and passed the 5% significance test. This indicates that the health status of the elderly tended to cluster spatially, with a good (poor) health province clustered near other good (poor) health provinces. According to Equation (2), a significant geographical agglomeration pattern of healthy elderly at the provincial-level could be identified as well (Figure 3). The very good health group formed a good health hot-spot region in the Yangtze River Delta, including Shanghai, Jiangsu and Zhejiang. In contrast, a poor health hot-spot region was formed in Inner Mongolia and Gansu in the north of China.

### 3.2. Determinants of Elderly Health Inequality in China

#### 3.2.1. Determinants of Inter-Provincial Health Inequality by Multi-Level Regression

We first constructed the null model to assess inter-provincial elderly health differences according to Equation (6). The elderly ill-health score variance $\sigma_{b}^{2}$ among provinces was 2.8203, and the intra-province variance $\sigma_{w}^{2}$ was 0.0694, ICC = 0.030, *p* < 0.0001. The ICC suggested that about 2.4% of the total variance in ill-health scores occurred among provinces. However, the results of the null model suggested that the development of a multi-level model was warranted, and we need to develop a multi-level model to explain this variability in intercepts within and among provinces.

According to the research method described in Section 2.5, provincial-level variables were added to Equation (6) to form Equation (7). Considering the multi-collinearity among these variables, we built eight regression models to test the impact of provincial-level factors on elderly health. Additionally, the interaction effects of air pollution and GDP per capita are added in Model 5 in order to reveal the complexity of the interaction between air pollution and elderly health according to the empirical study by Sun (2008) [38]. Table 3 shows that the level of medical, economic and educational resources had a more significant impact on elderly health at the provincial-level, as compared with the impact of environmental factors. The ill-health concentration curves are presented in Figure 4.

#### 3.2.2. Determinants for Health Inequality by Multi-Level Regression at the Individual Level

After screening out the three factors that significantly affected the elderly health shown in Model 1, 2 and 3, we inserted these three provincial level (level 2) factors into the model and include the individual variables identified by Equation (8). Using this method, we tested whether individual SES factors such as individual characteristics, social interaction, urban vs. rural locations, and income differences affected elderly health. The interaction effects of the place of residence (urban vs. rural) and GDP per capita at the provincial level were then added in Model 4 to reveal the complexity of the interaction among living place, inter-provincial economic difference and health status. After a collinearity examination, we find that the Variance Inflation Factor for each variable in all models were less than 1.5, and the multilevel regression models for testing the determinants of the ill-health scores are shown in Table 4. It can be seen that individual SES indicators such as social interaction, income, and living environment played a significant role in aggravating health inequality in China. The ill-health concentration curves of elderly health related to their location and income are shown in Figure 5.

## 4. Discussion

### 4.1. A Significant Geographical Differentiation in Elderly Health in China

According to the 4th SSEP, the overall elderly health status in China shows a “high in coastal, poor in central” spatial pattern [26,46,49] with the higher health provinces being Tianjin, Jiangsu, Shanghai, Zhejiang, Fujian and Beijing. Combined with the studies discussed in Section 3.2 that analyze the determinants of health differences, we confirm that these six most affluent provinces have the best economic, medical and educational conditions. Based on the analysis of Local Moran’s *I* cluster, Figure 3 shows that the high-health group formed a health hot-spot region in the Yangtze River Delta. Comparing our results with the conclusions of Du (2013) on the health inequalities of the elderly before 2008 [37], we find that the higher health hot-spot region was shrinking, and Jilin, Liaoning, Beijing, Tianjin, Hebei, Shanxi, Shandong, Henan, Hubei and Anhui provinces no longer belonged to the health hot-spot region. This indicates that inter-provincial health equality is increasing from a geographical perspective. In contrast, Inner Mongolia and Gansu, which are places in which the SES in China is the worst, make up a poor health hot-spot region, indicating that this area and its neighboring provinces suffer the most serious health inequalities in China.

### 4.2. Elderly Health Inequality at the Provincial Level and Its Determinants

Health inequalities among social groups and regions are ubiquitous in most countries. Identifying the key mechanisms that underpin the uneven spatial distribution in health outcomes has emerged as an important domain of academic enquiry in public health and human geography [22,55]. In China, analysis of the determinants of elderly health inequality at the provincial level in Section 3.2 shows that about 2.4% of elderly health differences are caused by inter-provincial differences, and this reflects the complex nature of health determinants and impact mechanisms. According to Equation (6), three provincial-level factors—“Grade-A tertiary hospitals per capita,” “Proportion of higher education,” and “GDP per capita”—significantly affected elderly health. This means that older people who lived in a region with a better economy, healthcare and education tended to have a better health status, resulting in significant geographical differentiation in elderly health in China.

“GDP per capita” is one of the core indicators of economic strength and closely related to residents’ quality of life. Similarly, “Grade-A tertiary hospitals per capita” is another core indicator of medical and health resources, and reflects the level of medical technology and health care service. Therefore, they are two critical factors of elderly health inequality [49]. Moreover, “Proportion of higher education population to total population” is also related to elderly health, given that well–educated people pay more attention to health than those less educated. However, unlike present studies [22,38,55,56,57,58], we find that environmental factors have less effect on spatial patterns of elderly health in China. Moreover, it is notable that the air quality variable is not the major cause of health inequalities in the provincial-level. For instance, in the very poor health provinces of China, Gansu has the worst air quality, and on the contrary, Hainan has the best air quality. Another interesting point is that the air quality in the very good health provinces is only rated as average (see Figure 1d and Figure 2). This is different from the study of Sun (2008) that regards air pollution as having a great effect on health status, and Chinese elderly who live in more developed areas are more susceptible to the effects of air pollution [38] based on a survey of 7358 elderly people.

In general, the main reason for inter-provincial differences in elderly health is the SES inequality in China. This is due to an imbalance in resource inputs and an unequal distribution of health-related resources such as health care and education in the disadvantaged regions, resulting in elderly health inequalities. Chinese policy makers need to re-orient their efforts toward “equalization” in the future medical system reforms, and it is necessary to make differential financial resource allocation plan according to the health needs of each area. That is, priority should be given to areas with high levels of medical and educational deprivation in order to promote equity in the distribution of resources and promote health equity.

### 4.3. Health Inequality of the Elderly at the Individual Level and Its Determinants

About 97.6% of China's elderly health inequalities are not caused by inter-provincial differences, but by factors such as age, gender, culture, marriage, social interaction and frequency of exercise at the individual level; this finding is consistent with many previous studies [59]. In addition, we find that the elderly of the Han ethnicity may have better health than other ethnic groups. This may be because the average socio-economic level of ethnic minorities is worse than that of ethnic Hans. Most ethnic minorities live in the central and western regions of China, which are less developed and have fewer medical resources [23,60].

The individual economic state is the main determinant of elderly health inequality, contributing to the most AIC components in model 5 (Figure 4, Table 4). Older people who have higher income and purchased commercial and social insurance always have a better health status. Since the elderly who purchased commercial and social insurance have a better health status, it is beneficial for elderly health to increase insurance coverage for people with low incomes [34]. The living environment is also an important factor of elderly health. Compared with elderly living in urban areas, the health status of residents closer to rural areas is much poorer, indicating an urban vs. rural-related inequality in elderly health (Figure 4). This is because many elderly people in rural areas cannot access regular medical treatments such as routine medical examinations. In contrast, elderly people in urban areas can have regular physical examinations, and they are able to arrive at the hospital in time for emergency treatments [61]. Additionally, in model 4, we find that the cross-level interaction variable *S3*
*× X10* shows a significant negative effect, indicating that the urban vs. rural-related elderly health inequality is larger in provinces where the economy is less developed. This is due to the fact that economically developed regions have a more balanced medical resource allocation. Moreover, the urban vs. rural-related inequality in underdeveloped regions may be more serious than we previously recognized and deserves more attention in the future.

In sum, we find that low-income elderly people—especially rural elderly people—in China tend to be in poor health. Pension insurance for China’s rural population was not launched until the 1990s. Until very recently, many low-income rural residents living in China did not pay pension insurance, which may explain why these residents typically do not have funds saved to use for medical treatment and cannot afford basic healthcare [62,63]. We believe that improving the current Chinese government health subsidy programs, promoting coordination between China's urban and rural medical resources and insurance, and encouraging rural residents to participate in insurance, especially in underdeveloped areas, may effectively reduce the health differences between urban and rural residents.

## 5. Conclusions

China is engaged in a rapid social and economic transition accompanied by rising economic disparities between the coast and inland, urban and rural areas, among provinces as well as within provinces. A large body of previous work has demonstrated that individual SES differences among the elderly in China do not entirely explain the inequities in elderly health at the inter-regional level. This article moves away from the analysis of individual-level factors and expands it to a geographic perspective, and our work reveals three main findings: (1) There were significant inter-provincial health differences among elderly people who live in the Yangtze River Delta region (better health) vs. those who live in Gansu and Hainan provinces (worse health). (2) Nearly 2.4% of the health differences in the elderly population in China were caused by inter-provincial differences—inequalities in economic, medical and educational resources drove these health inequalities. (3) At the individual level, inequalities of annual income have contributed to the disparity of the elderly health, and elderly people in less developed areas are more vulnerable to urban vs. rural-related health inequalities. Therefore, while it is impossible to reduce the economic gap among provinces within a short time period, promoting a more rational distribution of medical resources, increasing the coverage of health care facilities and strengthening the spread of health knowledge in underdeveloped regions may reduce health inequalities in the future.

Our study provides a reference to understand the relationship between public health differences and geographical determinants at the provincial and individual levels in China, and it suggests that regional conditions and SES inequalities should be taken into account while making elderly health policies. In future research, the use of continuous observation data of elderly health can help us explore factors that determine health inequality, such as health education, physical activity, social interaction and built environment. In general, future research should address the importance of health at the individual level as well as the locations in which people live, and policies that address health inequalities in the elderly population should require policy initiatives aimed at achieving greater regional equality in education and health care resources in China.

## Figures and Tables

**Figure 1 ijerph-16-02953-f001:**
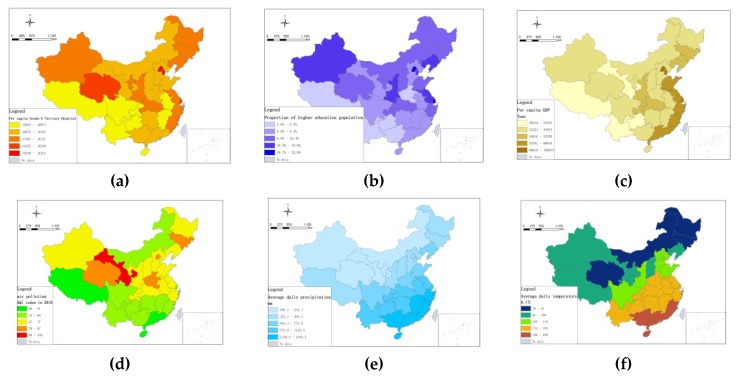
Inter-province differences for economic, medical resources, educational and environment in China. (a) Grade-A Tertiary Hospital Per capita; (b) proportion of higher education population to total population; (c) GDP per capita; (d) annual average air pollution index in 2010; (e) average daily precipitation; (f) average daily temperature; (g) average daily sunshine duration.

**Figure 2 ijerph-16-02953-f002:**
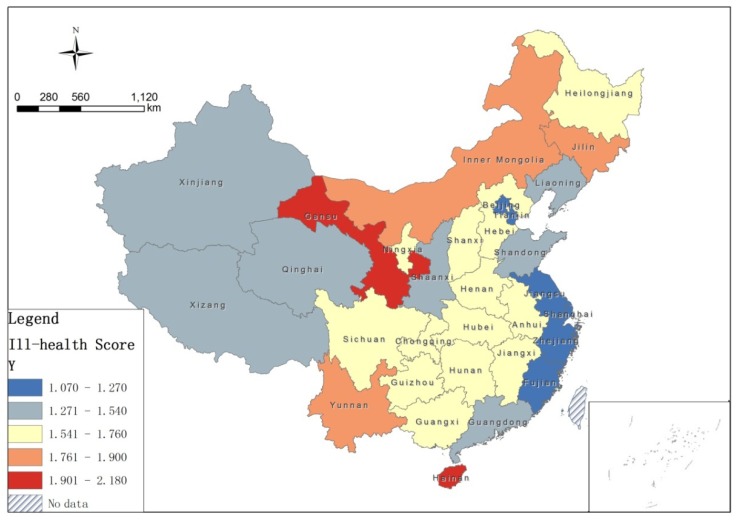
Ill-health score in 31 provinces of China.

**Figure 3 ijerph-16-02953-f003:**
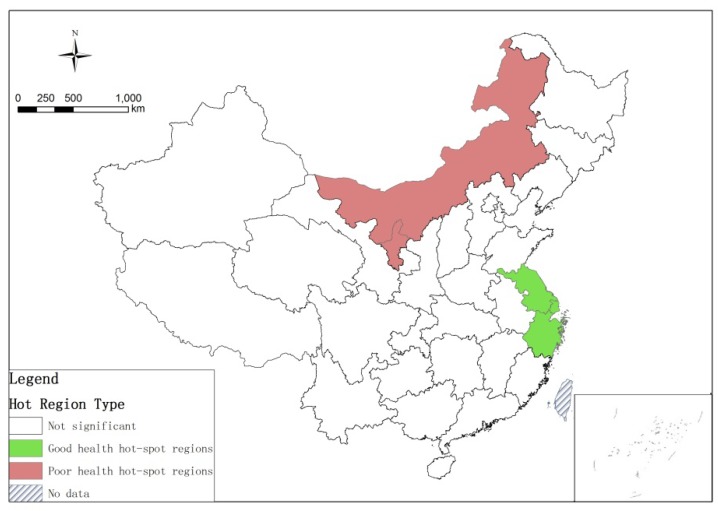
Local Moran's *I* cluster map of China's elderly health.

**Figure 4 ijerph-16-02953-f004:**
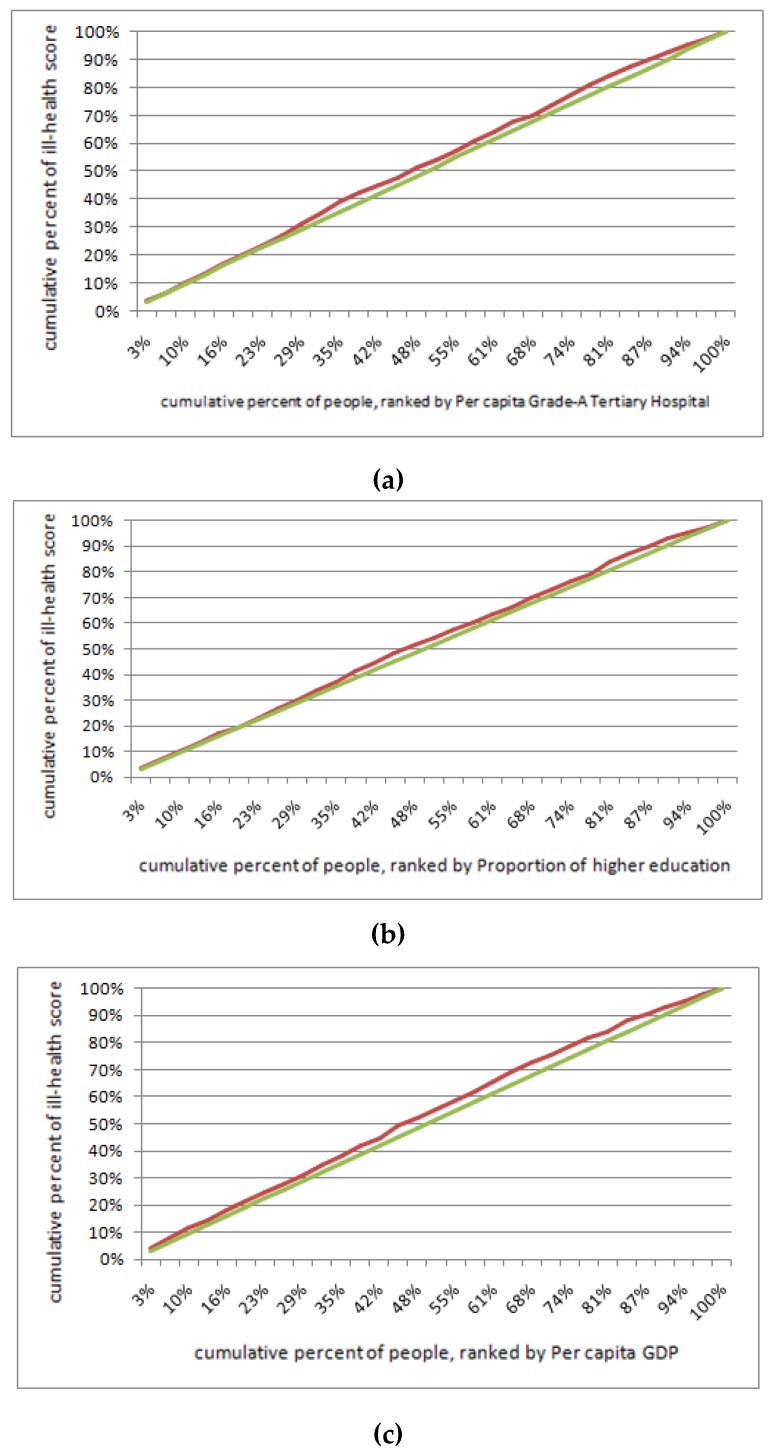
Ill-health concentration curve (red curve) of elderly health by socio-economic status (SES) difference at the provincial-level in China. (a) Medical resources-related elderly health inequality, *CI =* −0.083; (b) education-related elderly health inequality, *CI* = −0.093; (c) economic-related elderly health inequality, *CI* = −0.109.

**Figure 5 ijerph-16-02953-f005:**
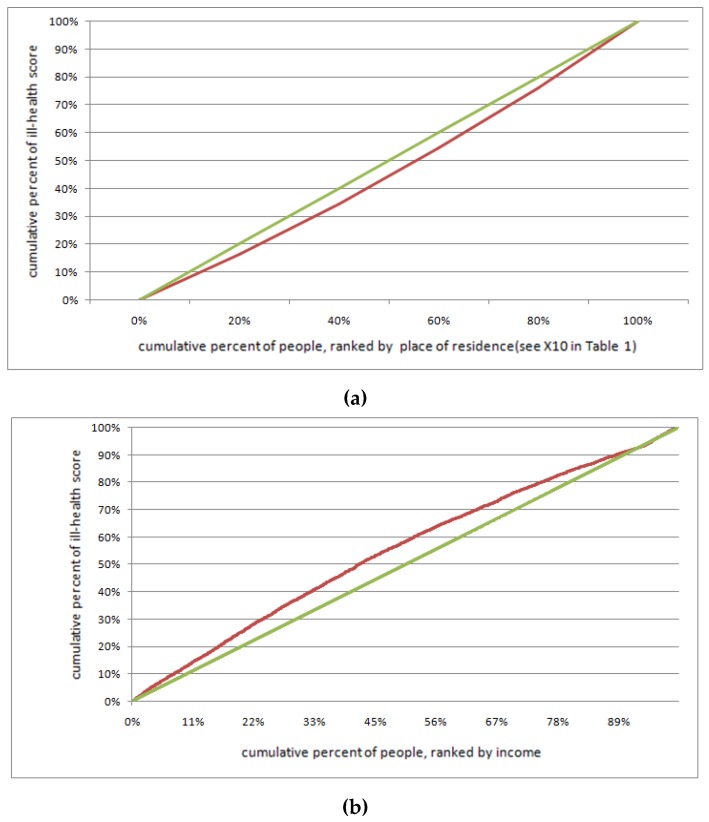
Ill-health concentration curve (red curve) of elderly health by SES difference at the individual level in China. (**a**) Urban vs. rural-related elderly health inequality, CI = 0.3544; (**b**) income-related elderly health inequality, CI = −0.1428.

**Table 1 ijerph-16-02953-t001:** Description of Variables and Expected Effects.

Level	Type		Code	Variable name	Expected effects on ill-health^1^	Calculation method
	Dependent variable		*Y*	Ill- health score		Ill-health score [47] (see 2.2)
Level 2: provincial-level	Explanatory variables	Medical resources [30,31]	*S_1_*	Grade-A tertiary hospital Per capita	-	Urban Statistical Yearbook of China
		Education [23]	*S_2_*	Proportion of higher education population to total population	-	The Sixth Population Census Data
		Economic	*S_3_*	GDP Per capita	-	Urban Statistical Yearbook of China
		Environment [36,41]	*S_4_*	Annual average air pollution index in 2010	+	Datacenter in Ministry of Environmental Protection of China
			*S* *_5_*	Average daily precipitation	?	Urban Statistical Yearbook of China, Unit: mm
			*S* *_6_*	Average daily temperature	+	Urban Statistical Yearbook of China, Unit: 0.1 °C
			*S* *_7_*	Average daily sunshine duration	-	Urban Statistical Yearbook of China, Unit: hours/year
Level 1: Individual-level	Control variables	Individual characteristics	*X_1_*	Age	+	
			*X_2_*	Gender	-	Male = 0, female = 1
			*X_3_*	Educational level	-	Uneducated = 1 Elementary school = 2 Junior high school = 3 High school/Technical secondary school/ Vocational high school = 4 College degree = 5 Bachelor degree or above = 6
			*X_4_*	Marital Status	-	Spouse is alive = 1Other = 0
			*X_5_*	Ethnicity	-	Han = 0, Other = 1
			*X_6_*	Exercise frequency	-	1 = No exercise 2 = Less than once in a week 3 = One to two times 4 = Three to Five Times 5 = Six times and above
		Social interaction	*X_7_*	Loneliness	-	Often = 1 Sometimes = 2 Never = 3
			*X_8_*	Social responsibility	-	Maintain community social security/Help mediate Neighborhood Disputes/Maintain Community Environment/Help Neighbors/Care For The Next Generation/ Participate In Cultural And Scientific Promotion Activities = 1 Do not participate = 0
			*X_9_*	Social activity	-	Watching movies / Dancing, Croquet/ Table tennis/ Badminton, Playing mahjong/Playing poker/Playing chess, Fishing/Calligraphy/ Photography/ Collection = 1 Do not participate = 0
	Explanatory variables	Built -environment	*X_10_*	Place of residence	+	Urban = 0, Urban-Rural area = 1, Town = 2, Town-Rural area = 3, Village = 4
			*X_11_*	House Type	+	Block = 1, Bungalow = 2, Mud house and other = 3
			*X_12_*	House quality	+	Property rights = 1 Lease and other = 2
		Personal economic situation	*X_13_*	Annual income	-	Ten thousand Yuan (Ln)
			*X_14_*	Social insurance	-	No social insurance = 0, else = 1
			*X_15_*	Commercial insurance	-	No commercial insurance = 0, else = 1

^1^ + positive correlation with ill-health score (positive for good health); - negative correlation with ill-health score; ? unknown effect on ill-health score.

**Table 2 ijerph-16-02953-t002:** Descriptive statistics of provincial-level and individual-level variables.

Variables		*N*	Min	Max	Average	Std.
*Y*	Ill-health score	221,518	0.14	7.61	1.5522	1.66329
*S_1_*	Grade-A tertiary hospital per capita	31	0.01	0.04	0.011	0.006
*S_2_*	Proportion of higher education population to total population	31	5.29	31.5	9.00	3.82
*S_3_*	GDP per capita	31	2.62	10.9	5.34	2.03
*S_4_*	Annual Average air pollution index (AQI) in 2010	31	37.86	108.91	69.12	9.48
*S* *_5_*	Average daily precipitation	31	100.20	1555.36	828.26	355.89
*S* *_6_*	Average daily temperature	31	15.78	247.84	144.59	44.23
*S* *_7_*	Average daily sunshine duration	31	834.63	2493.64	1676.14	363.49
Valid *N*		31				
*X_1_*	Age	222,179	60	109	69.731	7.84458
*X_2_*	Gender	222,179	0	1	0.4777	0.4995
*X_3_*	Educational level	221,445	1	6	2.1401	1.0509
*X_4_*	Marital Status	218,772	1	2	1.2791	0.44856
*X_5_*	Ethnicity	222,179	1	5	3.8798	1.75637
*X_6_*	Exercise frequency	220,903	1	5	2.5233	1.66817
*X_7_*	Loneliness	219,094	1	3	2.571	0.60939
*X_8_*	Social responsibility	215,366	0	1	0.46	0.498
*X_9_*	Social activity	215,706	0	1	0.92	0.27
*X_10_*	Place of residence	222179	1	5	3.5357	1.66677
*X_11_*	House Type	222179	1	2	1.0505	0.21896
*X_12_*	House quality	220777	1	3	1.6342	0.71889
*X_13_*	Annual income	218760	−5.3	12.21	0.8541	1.20279
*X_14_*	Social insurance	215395	0	1	0.0092	0.09541
*X_15_*	Commercial insurance	218067	0	1	0.038	0.19126
Valid *N*		210488				

**Table 3 ijerph-16-02953-t003:** Determinants for health inequality at the provincial level.

Variables name		Model 1	Model 2	Model 3	Model 4	Model 5	Model 6	Model 7	Model 8
		$\mathrm{Estimate}\;\gamma_{0j}$	$\mathrm{Estimate}\;\gamma_{0j}$	$\mathrm{Estimate}\;\gamma_{0j}$	$\mathrm{Estimate}\;\gamma_{0j}$	$\mathrm{Estimate}\;\gamma_{0j}$	$\mathrm{Estimate}\;\gamma_{0j}$	$\mathrm{Estimate}\;\gamma_{0j}$	$\mathrm{Estimate}\;\gamma_{0j}$
Intercept		1.829 ^***^	1.788 ^***^	1.996 ^***^					
*S_1_*	Grade-A tertiary hospital per capita	−0.027 ^**^							
*S_2_*	Proportion of higher education population to total population		−17.391 ^**^						
*S_3_*	GDP per capita			−0.803 ^***^		--			
*S_4_*	Annual average AQI				--	--			
*S_3_* *× S_4_*	GDP per capita× Annual average AQI					--			
*S* *_5_*	Average daily precipitation						--		
*S* *_6_*	Average daily temperature							--	
*S* *_7_*	Average daily sunshine duration								--

** *p* < 0.01; *** *p* < 0.001; – Not significant.

**Table 4 ijerph-16-02953-t004:** Determinants of health inequality.

Variables name			Model 1	Model 2	Model 3	Model 4	Model 5
			$\mathrm{Estimate}\;\gamma_{ij}$	$\mathrm{Estimate}\;\gamma_{ij}$	$\mathrm{Estimate}\;\gamma_{ij}$	$\mathrm{Estimate}\;\gamma_{ij}$	$\mathrm{Estimate}\;\gamma_{ij}$
	Intercept		0.173	3.000 ^***^	2.738 ^***^	2.683 ^***^	2.972 ^***^
Determinants for health inequality at the provincial level	*S_1_*	Grade-A tertiary hospital per capita	−0.911 ^*^	−0.411 ^*^	−0.330 ^*^	−1.118 ^*^	−0.551 ^*^
	*S_2_*	Proportion of higher education population to total population	−0.131 ^*^	−0.013 ^*^	−0.013 ^*^	−0.017 ^***^	−0.014 ^*^
	*S_3_*	GDP per capita	−0.411 ^***^	−0.050 ^***^	−0.049 ^***^	−0.060 ^***^	−0.052 ^***^
Individual characteristics at the individual level(Control variable)	*X_1_*	Age	0.030 ^***^	0.023 ^***^	0.030 ^***^	0.023 ^***^	0.023 ^***^
	*X_2_*	Gender	−0.166 ^***^	−0.174 ^***^	−0.182 ^***^	−0.182 ^***^	−0.173 ^***^
	*X_3_*	Educational level	−0.071 ^***^	−0.029 ^***^	−0.057 ^***^	−0.020 ^***^	−0.030 ^***^
	*X_4_*	Marital status	−0.013 ^*^	−0.302 ^***^	−0.013	−0.303 ^***^	−0.303 ^***^
	*X_5_*	Ethnicity	0.050 ^***^	0.037 ^***^	0.014 ^**^	0.009 ^*^	0.033 ^***^
	*X_6_*	Exercise frequency	−0.156 ^***^	−0.109 ^***^	−0.107 ^***^	−0.105 ^***^	−0.109 ^***^
Social interaction at the individual level (Control variable)	*X_7_*	Loneliness		−0.542 ^***^	−0.525 ^***^	−0.532 ^***^	−0.523 ^***^
	*X_8_*	Social responsibility		−0.211 ^***^	−0.206 ^***^	−0.214 ^***^	−0.201 ^***^
	*X_9_*	Social activity		−0.730 ^***^	−0.776 ^***^	−0.729 ^***^	−0.773 ^***^
Living Environment at the individual level	*X_10_*	Place of residence			0.022 ^***^	0.047 ^***^	
	*X_11_*	House type			0.086 ^***^	0.098 ^***^	
	*X_12_*	House quality			0.060 ^***^	0.062 ^***^	
	*X**_10_*×*S_3_*	Place of residence×GDP per capita				−0.004 ^***^	
Economy at the individual level	*X_13_*	Annual income					−0.027 ^***^
	*X_14_*	Social insurance					−0.207 ^***^
	*X_15_*	Commercial insurance					−0.088 ^***^
AIC			823615.9	767524.1	760488.6	762080.3	739731

* *p* < 0.05; ** *p* < 0.01; *** *p* < 0.001.

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
