# Peer review of "Elderly Health Inequality in China and its Determinants: A Geographical Perspective"

_ijerph, 2019, doi:10.3390/ijerph16162953_

Round 1

Reviewer 1 Report

See comments in manuscript.

Author Response

Response to Reviewer 1 Comments

Comment 1: On line 64Why? I would argue that the question has been very well discussed in many papers...

Response: Thank you for your valuable advice. Extensive research in this field has been carried out globally; however, due to a lack of data, health differences resulting from the unequal distribution of resources and environmental injustice have not been sufficiently discussed in China. We have clarified this in section 1.

Comment 2:On line 110, what does it mean “I separated the names, but this could be one source, is it?”

Response: We apologize for this mistake, and have revised it.

Comment 3:On line 341, is that a realistic recommendation? Does it make sens in the context?

Response: Thank you for your valuable advice. China's pension system was established late, and pension insurance for the rural population was only launched in the 1990s. Many low-income rural residents living in China did not pay pension insurance, which may be why they have no funds for medical treatment. We believe that improving the current system will help low-income, rural, elderly people to have access to medical care in this aging society. We have clarified this in section 4.3.

Reviewer 2 Report

Thank you for the opportunity to review this work. This paper aims to demonstrate the spatial patterns of elderly health inequality among Chinese provinces and its determinants. The paper would be scientifically strengthened if the authors made the paper more internationally appealing.

1. The authors could describe somewhere in the introduction: a) the use/value of geographic information systems in improving population’s health b) other studies that used spatial analysis to explore the spatial patterns of elderly health inequality.

2. Please could the authors clarify whether this was secondary analysis of data or whether the survey was conducted especially for the purpose of this study?

3. If possible, to explore the impact of area deprivation on the identified spatial patterns, it would strengthen the paper significantly.

4. The authors need to justify the use of said method. Moran's I is often considered a first -step test in spatial analysis, which is often followed by a more complicated spatial econometric method, e.g. spatial lag model/ geographic weighted regression. The authors should explain why multi-level linear regression model is enough to answer their research question.

5. Moran's I can also test if there is spatial correlation in the errors of two spatial polygons. As to whether or not such a correlation should be explained as a cluster or a spillover of health inequality is up for debate. Without a careful discussion of the reason of the observed significant Moran’s I, the conclusion is hard sale.

6. The government health subsidy (GHS) and healthcare utilization are important factors of elderly health. If possible, to add the impact of GHS and healthcare utilization, it would strengthen the paper significantly.

Some minor suggestions:

1. A more plausible definition of Local Moran's I will be nice.

2. Please could you revise the text in terms of language.

Author Response

Response to Reviewer 2 Comments

Comment 1:The authors could describe somewhere in the introduction: a) the use/value of geographic information systems in improving population’s health b) other studies that used spatial analysis to explore the spatial patterns of elderly health inequality.

Response: Geographical analysis is a very important tool in the study of spatial health inequalities. It is therefore necessary to use spatial analysis to explore the spatial pattern of elderly health differences. Thank you for your valuable advice. We have added a summary of the relevant literature in section 1.

Comment 2: Please could the authors clarify whether this was secondary analysis of data or whether the survey was conducted especially for the purpose of this study?

Response: Thank you for your advice. The goal of the survey was to investigate the living environments of elderly people in China and the needs of aged care services in different regions. We have clarified this in section 2.2.1.

Comment 3:If possible, to explore the impact of area deprivation on the identified spatial patterns, it would strengthen the paper significantly.

Response: Thank you for your valuable advice. The differences in the distribution of medical and educational resources discussed in this paper are types of inter-provincial deprivation in China. The conclusions also show that eliminating this deprivation contributes to the realization of health equity. We have added this to article 4.2, and have also highlighted policy recommendations.

Comment 4&5:The authors need to justify the use of said method. Moran's I is often considered a first -step test in spatial analysis, which is often followed by a more complicated spatial econometric method, e.g. spatial lag model/ geographic weighted regression. The authors should explain why multi-level linear regression model is enough to answer their research question. Moran's I can also test if there is spatial correlation in the errors of two spatial polygons. As to whether or not such a correlation should be explained as a cluster or a spill over of health inequality is up for debate. Without a careful discussion of the reason of the observed significant Moran’s I, the conclusion is hard sale.

Response: This paper uses the method of geographical differentiation to spatially explore the pattern of Chinese elderly health status. However, in the field of public health, research shows that human health differences are more influenced by individual conditions (such as behaviour, genetics, etc.). For example, in this paper, the ICC analysis of the health differences shows that more than 97% of health differences result from the individual-level differences discussed in section 3.1, rather than provincial-level differences. It is therefore difficult to explain health differences by using one single-level geographic unit weighted regression (e.g., spatial lag model/geographic weighted regression). In order to better explain the causes of health differences, the structure of the sample data must be considered to have a distinct two-level data hierarchy. Thank you for your valuable advice. We have explained this in the experimental design (see section 2.5), and the economic, medical, and education resource cluster in China is also discussed in section 4.1.

Comment 6: The government health subsidy (GHS) and healthcare utilization are important factors of elderly health. If possible, to add the impact of GHS and healthcare utilization, it would strengthen the paper significantly.

Response: Thank you for your valuable advice. China's pension system was established late, and pension insurance for the rural population was only launched in the 1990s. Therefore, many low-income, rural residents living in China did not pay pension insurance, which may be why they have no funds for medical treatment. Also, Chinese government health subsidy (GHS) programs to healthcare providers depend on the volume of healthcare services received by their patients. Socioeconomically disadvantaged population groups, such as many low-income and rural residents living in China, used healthcare less often than individuals with higher socioeconomic status, and received less subsidies than the advantaged population group, this cycle leads to an endemic inequality in access to healthcare when they are old. We believe that improving the current system will help low-income seniors to have access to medical care in this aging society. We have added this information in section 4.3.

Comment 7: Some minor suggestions: 1)A more plausible definition of Local Moran's I will be nice. 2)Please could you revise the text in terms of language.

Response: Thank you for your valuable suggestions. We have added an introduction to the Local Moran's I index in section 2.3 and revised the article.

Reviewer 3 Report

The article entitled “Elderly health inequality in China and its determinants: a geographical perspective” presents a cross-sectional study of the spatial differentiation of health inequality among Chinese provinces using a data that pertains to the elderly population in China. This is a very well written paper with thorough and complex statistical analyses. I have only a few minor edits to suggest for this paper:

1.       On line 50, what does it mean “(could you include a few of those areas here)”?

2.       This paper is based on a survey study and as such, it is a cross section study. Therefore any claim to causality should be eliminated. Such claims are made on line 92 and on line 368.

3.       Please include the corresponding references on lines 103-104 after the Declaration of Helsinki and also after the CRCA.

4.       On line 110, what does it mean “I separated the names, but this could be one source, is it?”

5.       In Table 1, please explain in the footnote the meaning of the symbols +, -, and ? in the column entitled “Expected Effects on Ill-health”

6.       On line 98 you say that your final study sample has the size of 222,179, but then in Table 2 your sample sizes vary for each of the variables considered in the analyses. Unless you imputed missing values, please remove the missing data from your Table 2, so that it reflects the complete data which has been used for your multivariable regression analyses models, Otherwise Table 2 can be very deceiving.

7.       Your Figure 4 and Figure 5 axes are almost impossible to read. Please enlarge the size of your figures to allow your reader to read your axes descriptions.

Author Response

Response to Reviewer 3 Comments

Comment 1:On line 50, what does it mean “(could you include a few of those areas here)”?

Response: Thank you for your valuable advice. We apologize for this mistake, and have revised it.

Comment 2:This paper is based on a survey study and as such, it is a cross section study. Therefore any claim to causality should be eliminated. Such claims are made on line 92 and on line 368.

Response: Thank you for your valuable advice. We have clarified this in sections 1 and 5.

Comment 3:Please include the corresponding references on lines 103-104 after the Declaration of Helsinki and also after the CRCA.

Response: Thank you for your valuable advice. We have carefully studied the content of our research. This article is a study of public health, and does not involve the Helsinki Declaration regarding medical content. Therefore, we deleted the relevant passage, as it was not essential to the content of the paper.

Comment 4:On line 110, what does it mean “I separated the names, but this could be one source, is it?”

Response: Thank you for your valuable advice. We apologize for this mistake, and have revised it.

Comment 5: In Table 1, please explain in the footnote the meaning of the symbols +, -, and ? in the column entitled “Expected Effects on Ill-health”

Response: Thank you for your valuable advice. The symbol “+” means a positive correlation with the ill-health score (positive for good health); the symbol “-” means a negative correlation with the ill-health score; the symbol “?” means an unknown effect on the ill-health score. We have explained this and added a footnote in Table 1.

Comment 6: On line 98 you say that your final study sample has the size of 222,179, but then in Table 2 your sample sizes vary for each of the variables considered in the analyses. Unless you imputed missing values, please remove the missing data from your Table 2, so that it reflects the complete data which has been used for your multivariable regression analyses models, Otherwise Table 2 can be very deceiving.

Response: Thank you for your valuable advice. We have added Valid N in Table 2.

Comment 7: Your Figure 4 and Figure 5 axes are almost impossible to read. Please enlarge the size of your figures to allow your reader to read your axes descriptions.

Response: Thank you for your valuable advice. We have enlarged the sizes of our figures and provided the high-resolution versions to the editor.

Reviewer 4 Report

This study examines the social and geographic determinants of health disparities among the elderly population in China. It addresses an interesting and meaningful research question, and the findings have significant implications for policy and future research.

However, there are several issues that need to be addressed to improve the quality of this paper.

Major issues:

1.       It is unclear how the latent continuous health outcome variable was constructed based on the 5-caterogy ordinal self-rated health outcome. More information needs to be provided on the statistical foundation for this treatment, in addition to the four references included. I think it would help the readers get a better sense of the rationale behind the methods to include a couple sentences on the essence of the methods.

2.       You examined level 2 factors such as average daily temperature and average daily sunshine duration. It’d be good to cite some references that suggested an association between these factors and health outcomes.

3.       In Table 2, the N for the level 2 variables should be the number of the level two units, i.e., the provinces, and not—or at least in addition to—the individuals.

4.       The maps in Figure 1 showing the provincial level statistics are very informative, and they set up the narrative of this study very well. However, they are very hard to read. The legends are barely readable. The authors are strongly suggested to enlarge the graphs.

5.       For Table 3,  you only included the level 2 predictors. What is the rationale for doing this? You also only included only one indicator in each model (model 1, 2, and 3). Why not include all indicators in the same module simultaneously?

6.       Since all three level 2 indicators were found to be significant predictors in Table 3, why not include all of them in Table 4?

7.       Furthermore, in Table 4, it’d be easier for the readers if you could do a better job distinguishing level 1 versus level 2 predictors. This would be particularly important if you are considering adding other level 2 predictors in the models.

Minor issues:

1.       Borders should be removed in all the tables. It is easier to read tables without solid cell borders.

2.       The manuscript would benefit language editing and proofreading. Grammatical errors such as dangling modifiers need to be corrected, e.g., “by collecting health status data and individual…” sentence in the abstract section, and “taking China as an example, the health status of many of its citizen…” Past tense should be used consistently throughout the manuscript.

3.       Formatting editing is also needed. Missing spaces in the Figure 1 title between sentences is just one of the examples. There are also significant discrepancie in fonts used and spacing options. In Figure 2 and 3, there’s a missing space in “NoData.” Details matter.

Author Response

Response to Reviewer 4 Comments

Comment 1:It is unclear how the latent continuous health outcome variable was constructed based on the 5-caterogy ordinal self-rated health outcome. More information needs to be provided on the statistical foundation for this treatment, in addition to the four references included. I think it would help the readers get a better sense of the rationale behind the methods to include a couple sentences on the essence of the methods.

Response: Thank you for your valuable advice. We have added more information about how the latent continuous health outcome variable was constructed based on the 5-category, ordinal, self-rated health outcome in section 2.2.

Comment 2:You examined level 2 factors such as average daily temperature and average daily sunshine duration. It’d be good to cite some references that suggested an association between these factors and health outcomes.

Response: Thank you for your valuable advice. In many previous climate-related articles, the effects of temperature, rainfall, and sunshine on the health of the elderly have been mentioned. We have added citations in section 2.1.1.

Comment 3:In Table 2, the N for the level 2 variables should be the number of the level two units, i.e., the provinces, and not—or at least in addition to—the individuals.

Response: Thank you for your valuable advice. We apologize for this mistake, and have revised it in Table 2.

Comment 4: The maps in Figure 1 showing the provincial level statistics are very informative, and they set up the narrative of this study very well. However, they are very hard to read. The legends are barely readable. The authors are strongly suggested to enlarge the graphs.

Response: Thank you for your valuable advice. We have enlarged the sizes of our figures and provided the high-resolution versions to the editor.

Comment 5: 1)For Table 3,  you only included the level 2 predictors. What is the rationale for doing this? 2) You also only included only one indicator in each model (model 1, 2, and 3). Why not include all indicators in the same module simultaneously?

Response: Thank you for your valuable advice. 1) In the 2-level model construction process, the first step is to determine whether the provincial factors (level 2) influence the model-dependent variable variance. In section 3.2.1, ICC = 0.030 and p < 0.0001 indicates that provincial factors have a significant impact on health, so it is necessary to determine which provincial factors (S1 to S7) will have an impact. Thus, the models in Table 3 are included to first validate the health effects of the provincial factors.

 2) Because some factors at the provincial level have significant collinearity (the most obvious examples are daily precipitation, sunshine, and temperature), all factors cannot be included together in one model. After screening out the three factors that significantly affect elderly health in models 1, 2, and 3, we put these three provincial-level (level 2) factors into the model with individual variables (see Table 4). We have clarified this in section 3.2.

Comment 6&7:Since all three level 2 indicators were found to be significant predictors in Table 3, why not include all of them in Table 4? Furthermore, in Table 4, it’d be easier for the readers if you could do a better job distinguishing level 1 versus level 2 predictors. This would be particularly important if you are considering adding other level 2 predictors in the models.

Response: To save space in the original version, we did not include the level 2 variables in Table 4, as it was not conducive to reading. In the revised version, we have moved the level 2 significant predictors from Table 3 to Table 4 so the readers can more easily distinguish the level 1 from the level 2 predictors. Thank you for your valuable advice.

Comment 8: Minor issues.1)Borders should be removed in all the tables. It is easier to read tables without solid cell borders. 2) The manuscript would benefit language editing and proofreading. Grammatical errors such as dangling modifiers need to be corrected, e.g., “by collecting health status data and individual…” sentence in the abstract section, and “taking China as an example, the health status of many of its citizen…” Past tense should be used consistently throughout the manuscript. 3)Formatting editing is also needed. Missing spaces in the Figure 1 title between sentences is just one of the examples. There are also significant discrepancie in fonts used and spacing options. In Figure 2 and 3, there’s a missing space in “NoData.” Details matter.

Response: We have re-edited these tables, adjusted the tense of the article, and re-adjusted the legend. Thank you for your suggestions.

Round 2

Reviewer 2 Report

The authors have greatly improved the manuscript since the last review. The response to the review is thorough.

Reviewer 4 Report

The authors responses are sufficient. I recommend the manuscript to be accepted in current forms, with minor editorial formatting and proofreading.